# Overexpression of *PSAT1* Gene is a Favorable Prognostic Marker in Lower-Grade Gliomas and Predicts a Favorable Outcome in Patients with *IDH1* Mutations and Chromosome 1p19q Codeletion

**DOI:** 10.3390/cancers12010013

**Published:** 2019-12-18

**Authors:** Shang-Pen Huang, Yung-Chieh Chan, Shang-Yu Huang, Yuan-Feng Lin

**Affiliations:** 1Department of Neurology, Po-Jen General Hospital, Taipei 105, Taiwan; neuro1471147@yahoo.com.tw; 2Genomics Research Center, Academia Sinica, Taipei 11529, Taiwan; yungchieh.c@gmail.com; 3Graduate Institute of Clinical Medicine, College of Medicine, Taipei Medical University, Taipei 110, Taiwan; 4Department of Law, School of Law, Ming Chuan University, Taipei 111, Taiwan; 5Department of Obstetrics and Gynecology, Chang Gung Memorial Hospital Linkou Medical Center, College of Medicine, Chang Gung University, Taoyuan 33305, Taiwan; einstone220@yahoo.com.tw; 6Cell Physiology and Molecular Image Research Center, Wan Fang Hospital, Taipei Medical University, Taipei 116, Taiwan

**Keywords:** *PSAT1*, lower-grade gliomas (LGGs), biomarker, prognosis, TCGA, CGGA, *IDH1* mutation, 1p19q codeletion

## Abstract

Patients with lower-grade gliomas (LGGs) have highly diverse clinical outcomes. Although histological features and molecular markers have been used to predict prognosis, the identification of new biomarkers for the accurate prediction of patient outcomes is still needed. The serine synthesis pathway (SSP) is important in cancer metabolism. There are three key regulators, including phosphoglycerate dehydrogenase (PHGDH), phosphoserine phosphatase (PSPH), and phosphoserine aminotransferase 1 (PSAT1), in SSP. However, their clinical importance in LGGs is still unknown. In this study, we used the bioinformatics tool in the Gene Expression Profiling Interactive Analysis (GEPIA) website to examine the prognostic significance of *PHGDH*, *PSPH*, and *PSAT1* genes in LGGs. *PSAT1* gene expression was then identified as a potential biomarker candidate for LGGs. Datasets from The Cancer Genome Atlas (TCGA) and the Chinese Glioma Genome Atlas (CGGA) were further used to explore the prognostic role of *PSAT1* gene. Our results demonstrated that *PSAT1* overexpression is a favorable prognostic marker of LGGs and significantly correlated with patient age ≤40, and a lower WHO histological grade, as well as mutations in *IDH1*, *TP53* and *ATRX*, but not with chromosome 1p19q codeletions. More importantly, LGG patients with isocitrate dehydrogenase 1 (IDH1) mutations, chromosome 1p19q codeletions, and *PSAT1* overexpression may have the best overall survival (five-year survival rate: 100%). Finally, we observed a coordinated biological reaction between *IDH1* mutations and *PSAT1* overexpression, and suggested overexpression of *PSAT1* might enhance the function of mutant IDH1 to promote a favorable outcome in LGG patients. In conclusion, our study confirmed the importance of identifying the overexpression of *PSAT1* as a favorable prognostic marker of LGGs, which may compensate for the limitation of *IDH1* mutations and chromosome 1p19q codeletion in the prognostication of LGGs.

## 1. Introduction

Glioma is a major aggressive type of malignant brain tumors [1]. Glioma shows histological features similar to those of glial cells such as astrocytes, oligodendrocytes and ependymal cells. The 2016 World Health Organization (WHO) classification of tumors of the central nervous system (CNS) classified diffuse gliomas by both histological and molecular features such as IDH1 (isocitrate dehydrogenase 1)-mutant and wild-type glioblastoma, *IDH1*-mutant and chromosome 1p/19q codeleted oligodendrogliomas, and other gliomas [2]. Glioblastoma multiforme (GBM), defined as grade IV glioma, is the most common glioma with the poorest prognosis in the adult population [3]. Lower-grade gliomas (LGGs) are less aggressive and slow-growing tumors, which are defined as grade II and III gliomas [4,5]. LGG patients have highly diverse clinical outcomes. Although *IDH1* mutations and chromosome 1p/19q codeletions have been identified as favorable prognostic biomarkers of LGGs, a new biomarker is still needed that could further predict the outcomes of LGG patients with *IDH1* mutations and chromosome 1p/19q codeletion.

Serine is a major one-carbon source in cancer cells, and contributes to amino acid metabolism, nucleotide synthesis and the generation of reducing agents such as NADPH [6,7]. Moreover, serine is a nonessential amino acid and is in high demand in the brain. However, the serine obtained from a normal diet is insufficient to recruit in the brain due to poor transportation across the blood–brain barrier (BBB) [8]. Patients with serine deficiency may have severe neurological syndromes, such as seizures, severe psychomotor retardation and congenital microcephaly. In cancer cells, the nutrient deprivation of glucose activates the serine synthesis pathway (SSP), which recruits nucleic acid synthesis and cell cycle progression, and increases the antioxidant capacity. Hence, serine is an important substrate of nucleotide and glutathione synthesis to facilitate cancer cell survival and proliferation [9].

There are three key regulators in the SSP: 3-phosphoglycerate dehydrogenase (PHGDH), phosphoserine aminotransferase 1 (PSAT1), and phosphoserine phosphatase (PSPH). PHGDH protein is significantly expressed in pancreatic tumor tissues compared to the adjacent normal tissues, and regulated cyclin B1 and cyclin D1, which are involved in cell proliferation and migration [10]. PHGDH overexpression has been found in breast cancer samples, especially in those with the estrogen receptor-negative (ER: negative) phenotype [11]. PSPH is responsible for removing phosphate to form serine. PSPH is a c-Myc-mediated enzyme and increases as hepatocellular carcinoma progression to a malignant clinical stage [12]. Moreover, overexpression of PSPH is also a poor prognostic marker in colorectal cancer [13].

PSAT1 is a pivotal enzyme that governs the production of two metabolites, serine and α-ketoglutarate (α-KG), which are involved in one carbon metabolism and the TCA cycle, respectively. PSAT1 was regarded as a poor prognostic marker in non-small cell lung cancer [14], esophageal squamous cell carcinoma [15], breast cancer [16] and colorectal cancer [17] and contributes to cancer cell proliferation and metastasis. In contrast, maintenance of PSAT1 protein at a high level has been identified as a marker for a favorable outcome in GBM with regorafenib treatment [18].

Although PHGDH, PSAT1 and PSPH have been identified as prognostic biomarkers of a variety of cancers, including GBMs, if they can serve as potential biomarkers of LGGs has not been extensively studied before. Due to the lack of studies exploring the role of *PHGDH*, *PSAT1*, and *PSPH* genes in the prognostication of LGGs, we utilized a bioinformatics tool (GEPIA) to compare their gene expression profiles in LGGs, other tumors and normal tissues. We finally chose *PSAT1* as a biomarker candidate and carefully examined its prognostic significance in LGG patients using The Cancer Genome Atlas (TCGA) LGG dataset. The prognostic significance of *PSAT1* in LGGs was further validated using Chinese Glioma Genome Atlas (CGGA) dataset and the REMBRANDT (Repository for Molecular Brain Neoplasia Data) cohort. Our results confirmed that overexpression of *PSAT1* is a potential biomarker for a favorable outcome in LGG patients. More importantly, our results also demonstrated that LGG patients with *IDH1* mutations, chromosome 1p/19q codeletion and overexpression of *PSAT1* could have the best overall survival of all patients.

## 2. Results

### 2.1. PHGDH, PSPH, and PSAT1 Are Important Regulators of the SSP and Have Certain Prognostic Significance in Various Cancers

PHGDH, PSPH and PSAT1 are important regulators of the SSP. From a literature review, overexpression of PHGDH, PSPH and PSAT1 proteins are known to lead to poor outcomes in various cancers (Table 1). The first serine-synthetic enzyme PHGDH was reported to promote pancreatic [10] and breast cancer [11]. Overexpression of PSPH was found in hepatocellular cancer [12] and colorectal cancer [13]. PSAT1 mediates carbon metabolism and the TCA cycle, which provides energy and increases biomass in non-small cell lung cancer (NSCLC) [14], breast cancer [16] and colorectal cancer [17]. However, the prognostic roles of these three genes in LGGs are still unknown.

### 2.2. PSAT1 Is Highly Expressed and Significantly Prognostic in Lower-Grade Gliomas and Could Be a Potential Biomarker Candidate

To explore the expression status of the *PHGDH*, *PSAT1*, and *PSPH* genes in various cancers, a bioinformatics tool website was used in this study. The expression levels of *PHGDH*, *PSAT1* and *PSPH* in tumors and normal tissues were explored using the bioinformatics website GEPIA (http://gepia.cancer-pku.cn). The results showed that *PSAT1* was highly expressed in gliomas, including LGGs and GBMs. More interestingly, its expression in LGGs was the highest among various cancers (Figure 1A). The *PHGDH* gene was also highly expressed in LGGs (Appendix A), but *PSPH* was highly expressed in GBMs, but not in LGGs (Appendix A).

The expression levels of *PSAT1* were significantly higher in tumors (LGGs and GBMs) than in normal tissues (Figure 1B). The expression levels of *PHGDH* were significantly higher in LGGs than in normal tissues, but not significantly higher in GBMs than in normal tissues (Appendix A). The expression levels of *PSPH* were significantly higher in tumors (LGGs and GBMs) than in normal tissues, but the significance was more prominent in GBMs than in LGGs (Appendix A). When comparing the prognostic significance of *PHGDH*, *PSPH* and *PSAT1* genes in the TCGA LGG cohort, patients were classified into two groups according to the expression levels of above genes. Patients with a high expression of *PSAT1* had significantly better overall survival (OS) (median survival: 8.767 years; five-year survival rate: 71.1%) than those with a low expression of *PSAT1* (median survival: 5.247 years; five-year survival rate: 53.1%) (Figure 1C). However, the gene expression of *PHGDH* and *PSPH* had no prognostic significance in LGGs (Appendix A). Therefore, *PSAT1* rather than *PHGDH* and *PSPH* was chosen as a biomarker candidate for further analyses.

### 2.3. Overexpression of PSAT1 Correlates with Mutations in IDH1, ATRX and TP53 and a Lower Grade of LGGs, and Is Enriched in IDH1-Mutant LGGs without 1p19q Codeletion

The 2016 WHO classification of tumors of the CNS suggests the use of *IDH1* mutation status and chromosome 1p19q codeletion status to predict the prognosis of LGG patients [12]. According to the guidelines of the 2016 WHO classification of CNS tumors, patients in the TCGA LGG cohort (*n* = 520) were classified into three groups for further analyses, including *IDH1* wild-type, *IDH1* mutations with 1p19q codeletion, and *IDH1* mutations without 1p19q codeletion (Figure 2A). *PSAT1* appeared to be highly expressed in the group that had mutations in *IDH1* without 1p19q codeletion. Consistent with previously published literature [19,20,21], mutations in *TP53* and *ATRX*, as well as low expression of the *TERT* gene, were also enriched in the *IDH1* mutations without 1p19q codeletion group (Figure 2A).

The expression levels of *PSAT1* were significantly higher in the group of LGGs with *IDH1* mutations, with 1p19q noncodeletion, with *ATRX* mutations, with *TP53* mutations, with wild-type *CIC*, and with wild-type *FUBP1* (Figure 2B). When LGG patients were classified into three groups according to the *IDH1* mutation status and chromosome 1p19q codeletion status, the expression levels of *PSAT1* were significantly higher in the group of LGGs with *IDH1* mutations without chromosome 1p19q codeletion (Figure 2C).

In addition, the expression level of *PSAT1* was significantly and negatively correlated with that of *TERT* (Figure 2D). *IDH1* mutations are known to be a favorable prognostic marker of LGGs and more enriched in grade II gliomas compared to grade III gliomas. Similarly, the expression levels of *PSAT1* were also shown to be significantly higher in grade II gliomas than in grade III gliomas (Figure 2E). Moreover, the expression levels of *PSAT1* were significantly and negatively correlated with patient age (Figure 2F) and significantly higher in LGG patients with an alive status than in those with a dead status (Figure 2G).

Our results confirmed that the overexpression of the *PSAT1* gene correlates with mutations in *IDH1*, *ATRX* and *TP53*, a lower WHO grade of LGGs, as well as wild-type *CIC* and wild-type *FUBP1*, but not with chromosome 1p19q codeletion. In addition, overexpression of *PSAT1*, as well as mutations in *ATRX* and *TP53*, and low expression of *TERT* are enriched in *IDH1*-mutant LGGs without 1p19q codeletion.

### 2.4. PSAT1 Expression Is Significantly Prognostic in IDH1-Mutant, 1p19q Codeleted or 1p19q Not-Codeleted LGGs and LGGs Patients with IDH1 Mutations, Chromosome 1p19q Codeletion and Overexpression of PSAT1 Have the Best Overall Survival

*IDH1* mutation status and chromosome 1p19q codeletion status are the most important markers in LGGs. When LGG patients in the TCGA dataset (*n* = 520) were classified into two groups according to *IDH1* mutation status, the group with *IDH1* mutations (*n* = 404) had better OS (median survival: 8.186 years; five-year survival rate: 69.3%) than the group of *IDH1* wild-type (*n* = 116) (median survival: 2.123 years; five-year survival rate: 32.9%) (Appendix A). In addition, the prognostic value of 1p19q codeletion status in LGGs was also shown to be highly significant (Appendix A). When LGG patients were classified into four groups according to *IDH1* mutation status and *PSAT1* expression status, high expression of *PSAT1* was shown to be a significantly favorable prognostic marker in *IDH1*-mutant LGGs, but not in *IDH1* wild-type LGGs (Figure 3A). When LGG patients were classified into four groups according to chromosome 1p19q codeletion status and *PSAT1* expression status, high expression of *PSAT1* was shown to be a significantly favorable prognostic marker both in 1p19q codeleted and 1p19q not-codeleted LGGs (Figure 3B). When LGG patients were classified into three groups (*IDH1* wild-type, *IDH1* mutations with 1p19q codeletion, and *IDH1* mutations without 1p19q codeletion) according to the guideline of the 2016 WHO classification of CNS tumors, the group of *IDH1* wild-type (*n* = 116) had the worst OS (median survival: 2.123 years; five-year survival rate: 32.9%), and the group with *IDH1* mutations and chromosome 1p19q codeletion (*n* = 155) had the best OS (median survival: 12.863 years; five-year survival rate: 78.2%) (Figure 3C).

To provide a more accurate prognosis prediction of LGGs, the expression status of the *PSAT1* gene was incorporated with *IDH1* mutation status and chromosome 1p19q codeletion status to stratify LGG patients into six subgroups (Figure 3D). When *PSAT1* expression was incorporated, LGG patients with *IDH1* mutations and chromosome 1p19q codeletion (*n* = 155) were significantly stratified into two clinically distinct subgroups depending on *PSAT1* expression (*p* = 0.004628). LGG patients with *IDH1* mutations, chromosome 1p19q codeletion, and a high expression of *PSAT1* (*n* = 66) had significantly better OS (median survival: 12.863 years; five-year survival rate: 100%) than those with a low expression of *PSAT1* (*n* = 89) (median survival: 7.964 years; five-year survival rate: 67.0%). In addition, by incorporating *PSAT1* expression, LGG patients with *IDH1* mutations but not chromosome 1p19q codeletion (*n* = 249) were stratified into two distinct subgroups significantly (*p* = 0.033793). LGG patients with *IDH1* mutations and a high expression of *PSAT1*, but not chromosome 1p19q codeletion (*n* = 154) had significantly better OS (median survival: 8.186 years; five-year survival rate: 70.5%) than those with a low expression of *PSAT1* (*n* = 95) (median survival: 5.296 years; five-year survival rate: 59.1%) (Figure 3D). However, when LGG patients with wild-type *IDH1* (*n* = 116) were stratified into two subgroups depending on *PSAT1* expression, the difference in OS between the two subgroups was not statistically significant (five-year survival rate: 41.5% vs. 27.5%, *p* = 0.720566) (Figure 3D).

Since the group with *IDH1* mutations and a high expression of *PSAT1,* but not chromosome 1p19q codeletion (*n* = 154), had better OS (median survival: 8.186 years; five-year survival rate: 70.5%) than the group with *IDH1* mutations, chromosome 1p19q codeletion and a low expression of *PSAT1* (*n* = 89) (median survival: 7.964 years; five-year survival rate: 67.0%) (Figure 3D)*,* we then tried to set up a new algorithm by incorporating *PSAT1* expression with *IDH1* mutation status (Figure 3E). *PSAT1* expression significantly (*p* = 0.005027) separated LGG patients with *IDH1* mutations into more distinct subgroups (Figure 3A) compared to chromosome 1p19q codeletion status (*p* = 0.044243) (Figure 3C). Based on our results, we suggest a new algorithm for classifying LGG patients into more clinically relevant subgroups depending on *IDH1* mutation status, the expression status of *PSAT1* and chromosome 1p19q codeletion status (Figure 3E).

Our results suggested that gene expression of *PSAT1* may be incorporated with *IDH1* mutation status and chromosome 1p19q codeletion status to provide a more accurate prognosis prediction in LGGs. Although *PSAT1* expression was relatively lower in LGGs with 1p19q codeletion, the stratification of combining *IDH1* mutations, chromosome 1p19q codeletion and *PSAT1* overexpression predicted the best OS in the TCGA LGG patients (median survival: 12.863 years; five-year survival rate: 100.0%; adjusted Hazard Ratio = 0.118) (Figure 3D,E).

### 2.5. Validating the Prognostic Significance of PSAT1 in Grade II and III Gliomas Using the CGGA Dataset

To validate the prognostic significance of the *PSAT1* gene, LGG (grade II and III gliomas) patients (*n* = 318) in the CGGA dataset were used as a validation cohort. In our results, the expression levels of *PSAT1* were also significantly higher in grade II and III gliomas with *IDH1* mutations than in those of *IDH1* wild-type (Figure 4A), and significantly higher in grade II gliomas than in grade III gliomas (Figure 4B). However, the expression levels of *PSAT1* were not significantly distinct between 1p19q codeleted and 1p19q not-codeleted LGGs (Figure 4C). When LGG patients were classified into three groups (*IDH1* wild-type, *IDH1* mutations with 1p19q codeletion and *IDH1* mutations without 1p19q codeletion) according to the guideline of the 2016 WHO classification of CNS tumors, the expression levels of *PSAT1* were highest in the LGG group with *IDH1* mutations but not chromosome 1p19q codeletion. And the expression levels of *PSAT1* were significantly distinct among the 3 LGG groups (Figure 4D). When LGG patients in the CGGA dataset were classified into 4 groups according to the WHO grade and *IDH1* mutation status, the expression levels of *PSAT1* were highest in the group of grade II gliomas with *IDH1* mutations, which was supposed to be the group with the best prognosis in LGGs (Figure 4E).

The prognostic significance of *IDH1* mutation status and 1p19q codeletion status in LGGs in the CGGA dataset were shown to be highly significant (Appendix A). When LGG patients in the CGGA dataset were classified into two groups according to *PSAT1* expression for survival analysis, patients with a high expression of *PSAT1* were shown to have significantly (*p* = 0.004216) better OS (median survival: 5.679 years; five-year survival rate: 54.3%; adjusted HR = 0.629) than those with a low expression of *PSAT1* (median survival: 3.652 years; five-year survival rate: 40.0%) (Figure 4F). When LGG patients were classified into three groups according to the guideline of the 2016 WHO classification of CNS tumors, patients with *IDH1* mutations and 1p19q codeletion had the best OS (median survival: 6.879 years; five-year survival rate: 77.3%; adjusted HR = 0.221), but those with wild-type *IDH1* had the worst OS (median survival: 2.003 years; five-year survival rate: 34.1%) (Figure 4G).

When LGG patients in the CGGA dataset were classified into 6 groups according to *IDH1* mutation status, 1p19q codeletion status and the expression status of *PSAT1*, the group with *IDH1* mutations, 1p19q codeletion and a high expression of *PSAT1* had the best OS (median survival: 6.879 years; five-year survival rate: 93.9%; adjusted HR = 0.189), but the group with wild-type *IDH1* and a high expression of *PSAT1* had the worst OS (median survival: 1.836 years; five-year survival rate: 30.3%; adjusted HR = 2.009) (Figure 4H). Overexpression of *PSAT1* was shown to be a significantly favorable prognostic marker in *IDH1*-mutant LGGs with 1p19q codeletion (*p* = 0.006988) as well as in *IDH1*-mutant LGGs without 1p19q codeletion (*p* = 0.008510) in the CGGA cohort (Figure 4H).

Our results confirmed that overexpression of the *PSAT1* gene correlates with *IDH1* mutations, a lower tumor grade, and a better outcome in LGGs in the CGGA dataset. Furthermore, LGG patients with *IDH1* mutations, 1p19q codeletion and overexpression of *PSAT1* had the best prognosis of all patients. Another LGG cohort (the REMBRANT cohort, *n* = 329) was used for further validation and the result revealed that overexpression of *PSAT1* is a significantly favorable prognostic marker of LGGs (Appendix A).

## 3. Discussion

Our study is the first to identify overexpression of the *PSAT1* gene as a favorable prognostic marker of LGGs. In our results, we demonstrated that overexpression of *PSAT1* predicted a favorable outcome of LGG patients in the TCGA dataset (Figure 1C). In addition, overexpression of the *PSAT1* gene is significantly correlated with alive patient status, patient age ≤ 40, a lower WHO histological grade, *IDH1* mutations, *TP53* mutations, *ATRX* mutations, wild-type *FUBP1*, low expression of *TERT* and chromosome 1p19q noncodeletion, but not with wild-type *CIC* (Table 2). The correlations of *PSAT1* overexpression with *IDH1* mutations, a lower WHO grade and a favorable outcome of LGG patients were also validated in the CGGA dataset (Figure 4A,B,F) and the REMBRANT cohort (Appendix A). The correlations between the promoter methylation status of *PSAT1* and other parameters in LGGs were also assessed Appendix A). There was a significant correlation between the promoter methylation status and gene expression of *PSAT1* (Appendix A). There were also significant correlations between the promoter methylation status of *PSAT1* and other parameters (Appendix A). And the high promoter methylation status of the *PSAT1* gene was shown to be a significantly favorable prognostic marker in the TCGA LGG cohort (Appendix A).

Although the 2016 WHO classification of CNS tumors suggested using *IDH1* mutation status and chromosome 1p19q codeletion status to stratify LGG patients into clinically distinct subgroups, this classification still remains limited in prognosis predictions. For example, even though *IDH1* mutations and chromosome 1p19q codeletions are both favorable prognostic markers, some LGG patients with those favorable prognostic markers still died rapidly. Due to the limitations of current biomarkers and classification tools, we identified *PSAT1* overexpression as a favorable prognostic marker, which may be incorporated with *IDH1* mutation status and chromosome 1p19q codeletion status to stratify LGG patients into more clinically relevant subgroups and provide a more accurate prognosis prediction in LGG patients. In this study, we demonstrated that in LGG patients with *IDH1* mutations and chromosome 1p19q codeletions, those with *PSAT1* overexpression may have significantly (*p* = 0.004628) better OS (median survival: 12.863 years; five-year survival rate: 100%) than those with low *PSAT1* expression (median survival: 7.964 years; five-year survival rate: 67.0%) (Figure 3D). Our study confirmed the importance of identifying the overexpression of *PSAT1* as a favorable prognostic marker in LGGs, which may compensate for the limitation of *IDH1* mutations and chromosome 1p19q codeletions in the prognostication of LGGs.

An underlying mechanism that explains how overexpression of the *PSAT1* gene contributes to a favorable outcome in LGG patients remains unclear. A number of studies have suggested overexpression of PSAT1 protein as a poor prognostic marker in colorectal cancer [17], esophageal squamous cell carcinoma [15], NSCLC [14] and ER(−) breast cancer [16]. However, a recent study suggested that a high level of PSAT1 protein could be a favorable prognostic marker for regorafenib-induced GBM suppression [18]. From literature review and our analyses, we conclude that the prognostic role of *PSAT1* overexpression in gliomas, including LGGs and GBM, might be different from that in other cancer types.

*IDH1* is a metabolic gene that encodes an enzyme named isocitrate dehydrogenase 1. This enzyme normally converts isocitrate to α-KG [22]. When *IDH1* is mutated, there will be loss of its normal enzymatic function, the production of α-KG. In addition, there will be gain of a new function, the production of R-2-hydroxyglutarate (R-2-HG) (Figure 5B). The mutation in the *IDH1* gene was first described in cancers by Sjöblom et al. [23] and further identified to have clinical impact on GBM by Parsons et al. [24]. To date, *IDH1* mutations have been identified in a number of cancer types, especially in gliomas and acute myelogenous leukemia (AML). *IDH1* mutation is the most well-known prognostic biomarker of LGGs. LGG patients with *IDH1* mutations will have a better prognosis and therapeutic response than those with wild-type *IDH1*. Mutations in the *IDH1* gene are commonly present in more than 70% of LGGs and secondary glioblastomas [25], which is consistent with our data (404/520 = 77.7%) (Figure 2B and Figure 3A). However, the definite mechanism by which *IDH1* mutations promote a favorable outcome in patients with LGGs is still not well elucidated.

In our results, we observed a strong and significant correlation between *IDH1* mutations and high expression of the *PSAT1* gene (Figure 2B and Figure 4A) (Table 2). More importantly, overexpression of *PSAT1* was shown to be a favorable prognostic marker specifically in *IDH1*-mutant LGGs, but not in *IDH1* wild-type LGGs (Figure 3A and Figure 4H). Based on our findings, we hypothesized that there might be a strong connection between *IDH1* mutations and *PSAT1* overexpression to promote a favorable outcome of LGG patients. From literature review, we noticed that in the serine synthesis pathway, PSAT1 converts glutamate to α-KG (Figure 5A). In the TCA cycle, wild-type IDH1 converts isocitrate to α-KG, which is the same with the product of PSAT1 in the serine synthesis pathway. Conversely, mutant IDH1 converts α-KG to R-2-hydroxyglutarate (R-2HG) (Figure 5B). This finding is compatible with our hypothesis that overexpression of the *PSAT1* gene increases the production of α-KG which could be the substrate for mutant IDH1 to convert NADPH to NADP+ and gain R-2HG (Figure 5B). The reduction of NADPH results in the less regeneration of reduced glutathione, which plays an important role in antioxidation in mammalian cells and probably promotes resistance to chemotherapy or radiation induced apoptosis in LGGs [26,27]. In this study, we observed a strong and significant correlation between *IDH1* mutations and high expression of *PSAT1*. In addition, overexpression of *PSAT1* was shown to be favorably prognostic only in *IDH1*-mutant LGGs, but not in *IDH1* wild-type LGGs. From the literature review, we noticed there might be a coordinated biological reaction between *IDH1* mutations and a high expression of *PSAT1.* Our results suggested that overexpression of the *PSAT1* gene contributes to a favorable outcome in patients with LGGs, which is probably related to therapeutic resistance induced by *IDH1* mutations, and overexpression of *PSAT1* promote α-KG synthesis, which could enhance the function of mutant IDH1.

In summary, our findings concluded that overexpression of the *PSAT1* gene could be a potential biomarker for a favorable outcome in patients with LGGs. And overexpression of *PSAT1* could be incorporated with *IDH1* mutations and chromosome 1p19q codeletion to classify LGG patients and predict those with the best overall survival. Our results also suggested the coordinated biological reaction between *IDH1* mutations and overexpression of *PSAT1*, which may contribute to a favorable outcome in patients with LGGs.

## 4. Materials and Methods

### 4.1. Clinical Data and Gene Expression Profiles of LGG Patients from the TCGA Website

The TCGA website (http://xena.ucsc.edu/welcome-to-ucsc-xena/) provides a LGG (grade II and III gliomas) data for free download. The clinicopathological data, including age at diagnosis, WHO grade, overall survival time, and survival status, of patients deposited in the TCGA LGG dataset (*n* = 520) was collected from the aforementioned website.

The gene expression profiles, such as gene expression levels, mutation status of specific genes (e.g., *IDH1, TP53, ATRX*, *CIC*, and *FUBP1*) and copy number variation in specific chromosomes, of the TCGA LGG cohort were also downloaded from the above TCGA website (Appendix A).

### 4.2. Clinical Data and Gene Expression Profiles of LGG Patients from the CGGA Website

The CGGA (Chinese Glioma Genome Atlas) dataset was used as a validation cohort. The clinical information and gene expression profiles of LGG patients in the CGGA dataset (*n* = 318), including *IDH1* mutation status, chromosome 1p19q codeletion status, the WHO histological grade, expression levels of *PSAT1*, overall survival time, and survival status were collected from the CGGA website (http://www.cgga.org.cn/) (Appendix A).

### 4.3. Classifying LGG Patients Into Distinct Subgroups for Further Analyses

LGG patients (*n* = 520) were divided into two distinct subgroups for further analyses according to age, WHO histological grade, survival status of patients, expression status of *PSAT1* and *TERT*, and the mutation status of *IDH1, TP53, ATRX*, *CIC*, and *FUBP1*. For the age factor, 40 years was determined as the cut-off age in this study because the age of 40 years separated LGG patients into two groups with similar numbers (Table 2). For gene expression of *PSAT1*, patients were ranked according to their expression levels of *PSAT1* and the median value of *PSAT1* expression levels was determined as the cut-off value to separate patients into two equal subgroups (*n* = 260).

### 4.4. Statistical Analyses

SPSS version 20.0 software (SPSS, Chicago, IL, USA) was employed to perform all statistical analyses. Pearson’s Chi-square test was used to analyze associations between the expression status of *PSAT1* and clinicopathological features. The median overall survival time, five-year survival rate, hazard ratio (HR) and survival curves were obtained and analyzed by using the Kaplan–Meier analysis, and differences in survival were determined by the log-rank test. The scatter plots and box plots were drawn using Prism 5 software (GraphPad Software Inc., San Diego, CA, USA) and Student’s *t*-Test was used to analyze differences in the gene expression levels of *PSAT1* between different subgroups of patients with LGGs. For all analyses, a *p*-Value of <0.05 was considered to be statistically significant. The symbols *, ** and *** denote *p* < 0.05, *p* < 0.01 and *p* < 0.001, respectively.

## 5. Conclusions

Our findings suggest that overexpression of the *PSAT1* gene severs as a favorable prognostic marker of LGGs, which could assist the limitation of *IDH1* mutations and chromosome 1p19q codeletion in the prognostication of LGGs clinically.

## Figures and Tables

**Figure 1 cancers-12-00013-f001:**
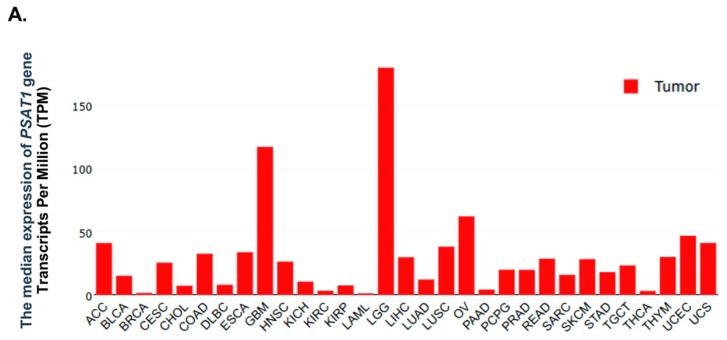
Identification of the *PSAT1* gene as a biomarker candidate of lower-grade gliomas. The expression status of *PSAT1* in tumors and normal tissues were obtained from the bioinformatics website, GEPIA. (**A**) The expression levels of *PSAT1* in various tumors. Gene levels of *PSAT1* were highly expressed in LGGs and GBMs. (**B**) The expression levels of *PSAT1* in LGGs, GBMs and normal tissues. The expression levels of *PSAT1* were significantly higher in tumors (LGGs and GBMs) than in normal tissues. (**C**) The prognostic significance of *PSAT1* expression in the TCGA LGG cohort. Patients were divided into two subgroups according to *PSAT1* expression for survival analysis.

**Figure 2 cancers-12-00013-f002:**
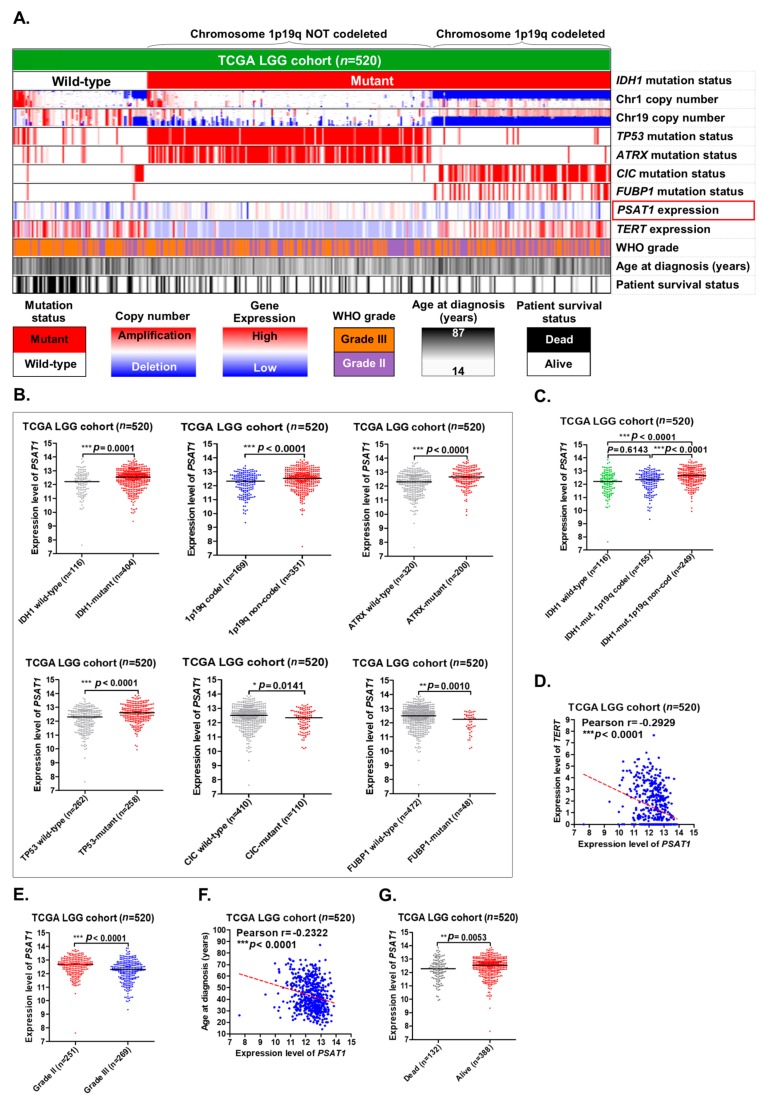
The correlations between *PSAT1* expression and other clinico-molecular parameters in LGGs in The Cancer Genome Atlas (TCGA) dataset. (**A**) A gene expression heatmap was constructed to show the correlations of *PSAT1* expression with other parameters in LGGs. LGG patients in the TCGA dataset (*n* = 520) were classified into three subgroups according to the 2016 World Health Organization (WHO) classification of central nervous system (CNS) tumors (*IDH1* wild-type, *IDH1* mutations with chromosome 1p19q codeletion, and *IDH1* mutations without chromosome 1p19q codeletion). (**B**) The correlations of *PSAT1* expression with *IDH1* mutations, chromosome 1p19q codeletion, *ATRX* mutations, *TP53* mutations, *CIC* mutations, and *FUBP1* mutations. (**C**) The expression levels of *PSAT1* in the three subgroups of LGGs classified according to the 2016 WHO classification of CNS tumors. (**D**) The correlation between *PSAT1* expression and *TERT* expression. (**E**) The expression levels of *PSAT1* in grade II and III gliomas. (**F**) The correlation between *PSAT1* expression and patient age. (**G**) The expression levels of *PSAT1* in dead and alive LGG patients. The symbols *, ** and *** denote *p* < 0.05, *p* < 0.01 and *p* < 0.001, respectively.

**Figure 3 cancers-12-00013-f003:**
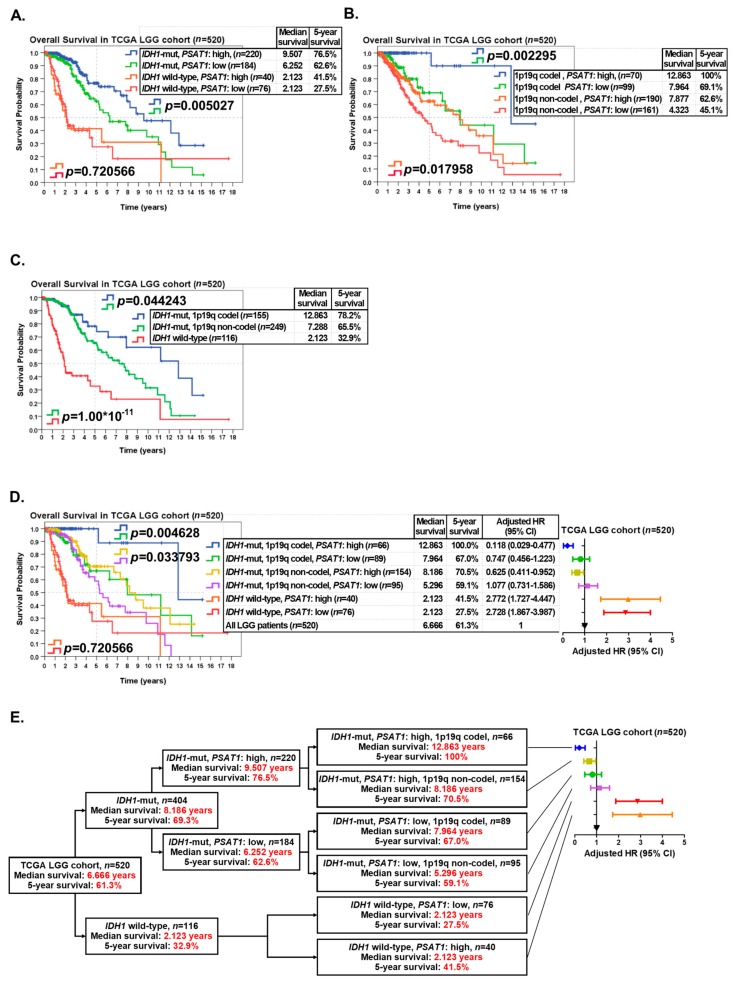
*PSAT1* expression status was incorporated with *IDH1* mutation status and chromosome 1p19q codeletion status to provide a better prognosis prediction in LGGs in the TCGA dataset. (**A**) The expression status of *PSAT1* was incorporated with *IDH1* mutation status to separate LGG patients into four subgroups for survival analysis. (**B**) The expression status of *PSAT1* was incorporated with chromosome 1p19q codeletion status to separate LGG patients into four subgroups for survival analysis. (**C**) LGG patients were divided into three subgroups according to *IDH1* mutation status and chromosome 1p19q codeletion status for survival analysis. (**D**) The expression status of *PSAT1* was incorporated with *IDH1* mutation status and chromosome 1p19q codeletion status to stratify LGG patients into six subgroups for survival analysis. (**E**) A suggested algorithm for classifying LGG patients with the combined use of *IDH1* mutation status, the expression status of *PSAT1* and chromosome 1p19q codeletion status.

**Figure 4 cancers-12-00013-f004:**
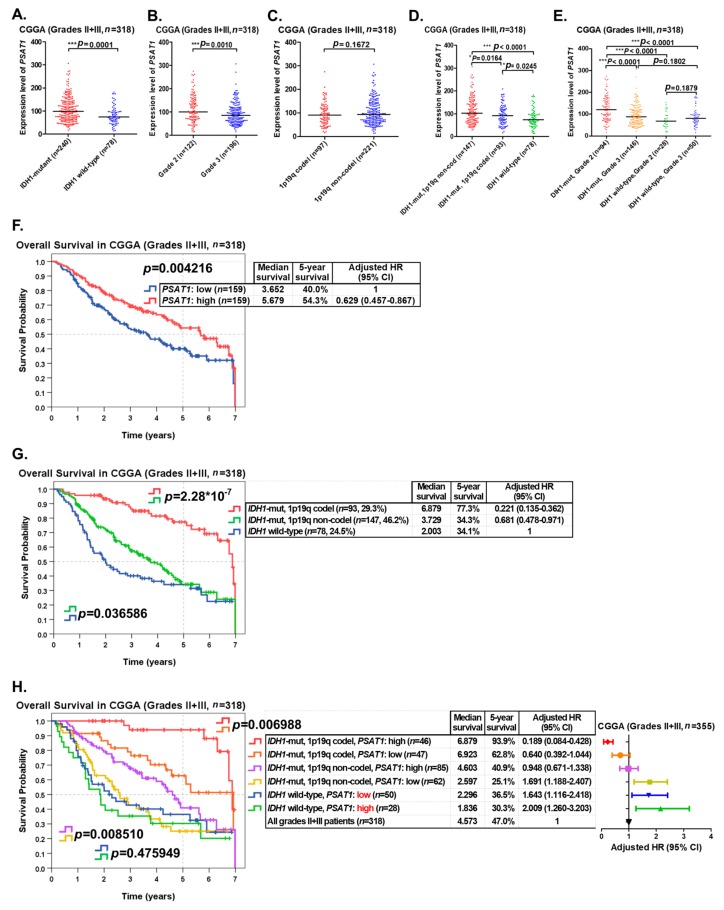
Validating the prognostic role of *PSAT1* overexpression in the CGGA dataset. LGG (grade II and III gliomas) patients (*n* = 318) in the CGGA (Chinese Glioma Genome Atlas) dataset were used as a validation cohort. (**A**) The expression levels of *PSAT1* in *IDH1*-mutant and *IDH1* wild-type LGGs. (**B**) The expression levels of *PSAT1* in grade II and grade III gliomas. (**C**) The expression levels of *PSAT1* in chromosome 1p19q codeleted and not-codeleted LGGs. (**D**) The expression levels of *PSAT1* in the three subgroups of LGGs classified according to the 2016 WHO classification of CNS tumors. (**E**) The expression levels of *PSAT1* in the four subgroups of LGGs classified according to *IDH1* mutation status and the WHO grade. (**F**) LGG patients were divided into two subgroups according to *PSAT1* expression fur survival analysis. (**G**) LGG patients were classified into three subgroups according to the guideline of the 2016 WHO classification of CNS tumors for survival analysis. (**H**) The expression status of *PSAT1* was incorporated with *IDH1* mutation status and 1p19q codeletion status to separate LGG patients into six subgroups for survival analysis. The symbols *, ** and *** denote *p* < 0.05, *p* < 0.01 and *p* < 0.001, respectively.

**Figure 5 cancers-12-00013-f005:**
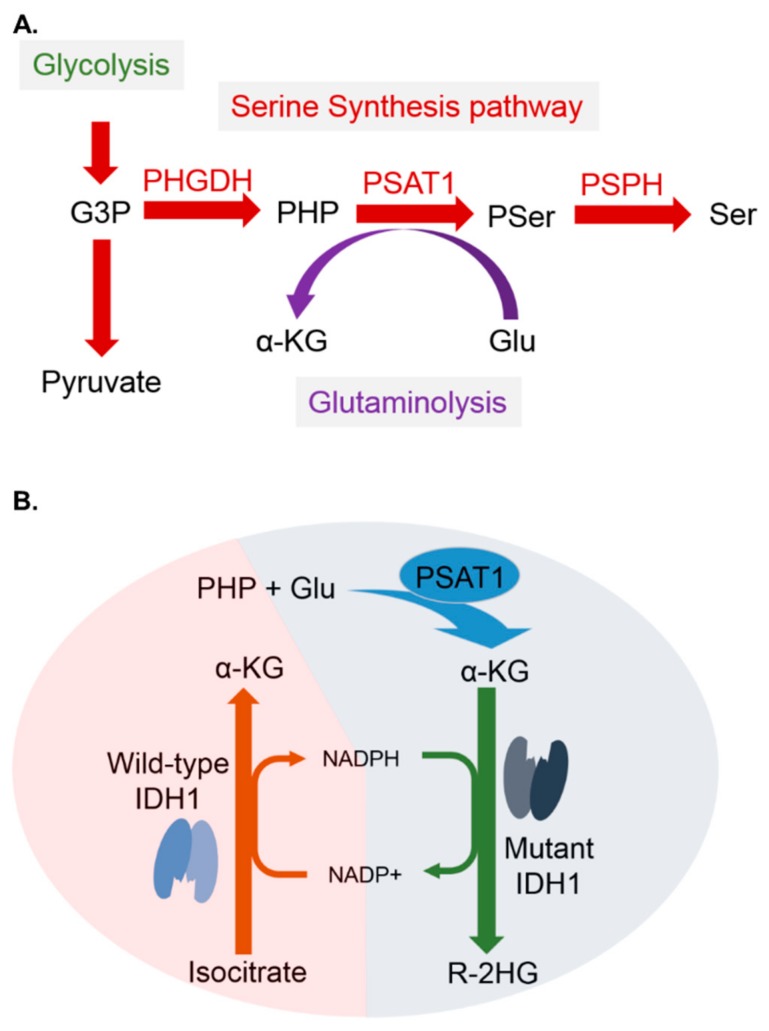
A coordinated biological reaction between *IDH1* mutations and *PSAT1* overexpression in LGGs. (**A**) *PSAT1* overexpression promotes α-KG synthesis. (**B**) Mutant IDH1 converts α-KG to R-2HG. *PSAT1* overexpression promotes α-KG synthesis, which could enhance the function of mutant IDH1.

**Table 1 cancers-12-00013-t001:** Key regulators in the serine synthesis pathway.

Enzyme	Classification	Substrate	Product	Prognostic Roles in Cancers
PHGDH	Oxidoreductase	3-phosphoglycerate; NAD^+^	3-phosphohydroxypyruvate; NADH; H^+^	A poor prognostic marker in Pancreatic cancer [10] and Breast cancer [11]
PSPH	Phosphatase	3-Phosphoserine	Serine; phosphate ion	A poor prognostic marker in Hepatocellular cancer [12] and Colorectal cancer [13]
PSAT1	Amino transferase	3-phosphohydroxypyruvate; glutamate	3-Phosphoserine; α-ketoglutarate	A poor prognostic marker in NSCLC [14], Esophageal Squamous Cell carcinoma [15], ER (-) Breast cancer [16] and Colorectal cancer [17] A favorable prognostic marker in Glioblastoma with regorafenib treatment [18]

NAD+: an oxidized nicotinamide adenine dinucleotide.

**Table 2 cancers-12-00013-t002:** Correlations of *PSAT1* expression with the clinicopathological features of patients in the TCGA LGG cohort.

Clinicopathological Feature	*N*	*PSAT1* Expression, N (%)		*p*
	520	Low, *n* = 260 (50.0)	High, *n* = 260 (50.0)	
Overall survival indicator				*** 0.000286
1 (dead)	132	84 (63.6)	48 (36.4)	
0 (alive)	388	176 (45.4)	212 (54.6)	
Chromosome 1p19q status				* 0.011473
Codeleted	169	98 (58.0)	71 (42.0)	
Not-codeleted	351	162 (46.2)	189 (53.8)	
Age				** 0.001172
>40	265	151 (57.0)	114 (43.0)	
≤40	255	109 (42.7)	146 (57.3)	
WHO histological Grade				*** 2.246 × 10^−7^
Grade III	269	164 (61.0)	105 (39.0)	
Grade II	251	96 (38.2)	155 (61.8)	
*IDH1* status				*** 0.000149
Wild-type	116	76 (65.5)	40 (34.5)	
Mutant	404	184 (45.5)	220 (54.5)	
*TP53* status				*** 0.000055
Wild-type	262	154 (58.8)	108 (41.2)	
Mutant	258	106 (41.1)	152 (58.9)	
*ATRX* status				*** 1.713 × 10^−7^
Wild-type	320	189 (59.1)	131 (40.9)	
Mutant	200	71 (35.5)	129 (64.5)	
*CIC* status				0.053261
Mutant	110	64 (58.2)	46 (41.8)	
Wild-type	410	196 (47.8)	214 (52.2)	
*FUBP1* status				** 0.006392
Mutant	48	33 (68.8)	15 (31.3)	
Wild-type	472	227 (48.1)	245 (51.9)	
*TERT* expression				*** 0.000026
High	260	154 (59.2)	106 (40.8)	
Low	260	106 (40.8)	154 (59.2)	

The symbols *, ** and *** denote *p* < 0.05, *p* < 0.01 and *p* < 0.001, respectively.

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
