# Peer review of "Overexpression of PSAT1 Gene is a Favorable Prognostic Marker in Lower-Grade Gliomas and Predicts a Favorable Outcome in Patients with IDH1 Mutations and Chromosome 1p19q Codeletion"

_cancers, 2019, doi:10.3390/cancers12010013_

Round 1
Reviewer 1 Report
The authors analyze TCGA and CGGA datasets, and identify a prognostic role for PSAT1 gene in IDH mutant gliomas.
Several of the presented results have already been described (Fig. 3A/B)
While TCGA LGG data is analyzed by molecular subgroups, CGGA cohort is not separated by 1p/19q status. The clinical information includes codeletion status and can be incorporated into the CGGA analysis.
IDH mutations are enriched in Grades II/III gliomas, not just Grade II glioma (lines 160-161)
If PSAT1 expression is prognostic in 1p/19q codel gliomas, it would be interesting to also assess correlation with recurrent mutations found in codels (e.g. CIC, FUBP1).
Is there a difference in promoter methylation status of PSAT1 gene within different IDH subgroups?
Typo in line 99...prognostic roles in LGGs are still known (should be 'unknown'?)
Reviewer 2 Report
Dear Authors,
Your manuscript on the relevance of PSAT1 gene expression in glioma is well designed and all used methods are state of the art. The findings are of interest for glioma research, but I would recommend to consider some points for revision:
To my impression, you put too much emphasis into the statistical analysis of your data. I do agree that the data you show is statistical significant different, but this is mainly due to the high number of cases. I do believe that the difference of tumor types (astrocytoma vs. oligodendroglioma) still has more impact than your marker gene expression, because this can be measured as a black and white pattern. The marker you describe is not so easy to measure and analyze. Or at least not with the data you give on this marker. So for establishing PSAT1 as prognostic marker, I would recommend to describe what exactly is meant with low and high expression. Proof this in a small independent cohort to show the prognostic significance of a single analysis of your marker. Best would be to translate this into something that can be used in daily routine diagnostic, like an antibody staining.Minor remarks:
Please ask a native speaker to review the manuscript. There are several minor grammar mistakes. The diagnosis oligoastrozytoma is not really valid anymore. I know that this diagnosis is still in the documentation, but based on the data you have for your analysis, I would only classify based on the criteria of the WHO classification of 2016. In my document several passages (e.g. line 181-182) seem to have a different text style. Please carefully check this in the manuscript. Line 298, the citation of Parson et al. is not correct. Parson was the first to show that the IDH mutations have a clinical impact, but not the first to describe them. They were show earlier, but in a too low incidence to have clinical significance (e.g. Sjöblom T. et al. Science 2006).Author Response
Please see the attachment.

Reviewer 3 Report
The authors present the article “Over-expression of PSAT1 gene is a favorable prognostic marker in lower grade gliomas and Predicts a Favorable Outcome of Patients withIDH1Mutations and Chromosome 1p19q Co-deletion.” In this article, their goal is to show that high PSAT expression is associated with improved overall survival and may be used as a biomarker for prognosis in specific patients with Grade II/III gliomas. My comments for the authors are as follows:
Overall Paper
It is very difficult to follow when this paper is referring to protein overexpression or gene overexpression. For example, line 93-94 says that over expression of PHGDH, PSAT1, PSPH genes are known to have poor outcomes in various cancers. But most of the papers referred to enzyme (protein) expression (as shown in the table). Gene expression does not always correlate with protein expression. Please make sure it is clear throughout the paper if the reference is gene or protein expression, and the methods employed to get this.Introduction
The sentence beginning on line 79 is not clear Please review the results of the Regorafenib study in GBMs. The conclusion of the paper does not align with the conclusion in this manuscriptResults
Figure 1B needs clearer labels- it is difficult to know which is the glioma and which is the normal tissue Paragraph 118-125; it is still unclear why PSAT1 over PHGDH and PSPH was chosen, especially since PHGDH also has significantly higher expression levels in LGGS than GBMs Line 135 and 136; this has been published before. Rather than saying "interestingly," it may be better to say “consistent with previously published literature” Figure 2- the legend for age is not clear (what does 87 and 14 mean?) The conclusion of 2.3 is that PSAT1 overexpression of PSAT1 correlates with IDH1, ATRX and TP53 mutations (which all occur in the same tumor), but not with chromosome 1p19q co-deletion (line 180-181), yet the conclusion of 2.4 is that IDH1 mutant tumors with 1p19q co-deletion and overexpression of PSAT1 have the best overall survival (line 225-226). Please explain the discrepancy between these two conclusions The primary conclusion of the TCGA cohort was that PSAT1+IDH1+1p19q co-deletion is the best overall response, yet the validation cohort did not stratify based on 1p19q or include this data at all. Please include this information Figure 4D and 4E- there is data missing in the tables Line 256-258: The validation cohort shows no difference in overall survival between PSAT1 high and PSAT1 low tumors, and does not show a survival difference between these groups even when adding IDH1. Therefore, the conclusion that LGG patients with IDH1 mutations and over-expression of PSAT1 had the best prognosis of all patients is not correct because the IDH1+PSAT1 low patients had similar prognosis.Discussion
Line 260-267; See comments above regarding specific comments about the data and conclusions Although interesting, there is no evidence to support the proposed mechanism. There are many other possible reasons the PSAT1 may be elevated in these tumors. Please provide additional evidence to support the proposed mechanism.Author Response
Please see the attachment.

Round 2
Reviewer 1 Report
The authors have addressed all my concerns.
Reviewer 2 Report
Dear Authors,
You have addressed all comments in an appropriate way and improved the manuscript substantially in respect to data presentation and language. I do think that your finding will have impact on future diagnostic, but I still miss a translation to a method suitable for daily diagnostic use. But this will hopefully be done in future.
Nevertheless I found one point you should check. The CGGA data seem to be compromised in the figures 4 and S3. The drop of the Kaplan Meyer plot at 7 years looks unnatural to me and it look ok in the first version of the manuscript. Please check this carefully.
Best regards
Reviewer 3 Report
Following revisions, the article "Over-expression of PSAT1 gene is a favorable prognostic marker in lower grade gliomas and predicts a favorable outcome of patients with IDH mutations and chromosome 1p19q co-deletion" is much improved. I have only a few minor suggestions to the authors.
Results
Page 3 line 95; the names of the protein do not need to be italicized
Line 140-142; this has been published before. Rather than saying Interestingly, it may be better to say “consistent with previously published literature.” In the response this is reported to have been corrected; but it was not corrected in the new manuscript that was submitted for re-review
Line 239; I believe this was intended to say "Although PSAT1 expression...the stratification of combining IDH1 mutations..."
It is still unclear what is meant by "cross-talk between PSAT1 mediated SSP and IDH1 mediated metabolic alteration." Is there a better way to write this so it is clear to the reader what this means?
